# On the Mortality of Companies

**DOI:** 10.3390/e24020208

**Published:** 2022-01-28

**Authors:** Peter Richmond, Bertrand M. Roehner

**Affiliations:** 1School of Physics, Trinity College Dublin, D02 PN40 Dublin 2, Ireland; 2Institute for Theoretical and High Energy Physics (LPTHE), Pierre and Marie Curie Campus, Sorbonne University, National Center for Scientific Research (CNRS), 75016 Paris, France; roehner@lpthe.jussieu.fr

**Keywords:** mortality, companies, start-up, FTSE100, Gompertz, MinMax, survival probability distribution

## Abstract

Using data from both the US and UK we examine the survival and mortality of companies in both the early stage or start-up and mature phases. The shape of the mortality curve is broadly similar to that of humans. Even small single cellular organisms such as rotifers have a similar shape. The mortality falls in the early stages in a hyperbolic manner until around 20–30 years when it begins to rise broadly according to the Gompertz exponential law. To explain in simple terms these features we adapt the MinMax model introduced by the authors elsewhere to explain the shape of the human mortality curve.

## 1. Introduction

In 1999, one of the authors of this paper (PR) arrived in Ireland to spend what became a decade in Trinity College. During that year he had the opportunity to attend the first ever European Physical Society sponsored conference on econophysics in Dublin. During the meeting he obtained a copy of the book ‘An Introduction to Econophysics’ by Rosario N Mantegna and H Eugene Stanley. As for many other physicists, that meeting and the book inspired new research directions. This paper is the latest in a series that have emerged from that initial revelation over two decades ago.

In a series of recent papers [1,2,3] the present authors have studied human mortality demonstrating how the shape of the mortality function has a bathtub type of shape where the infant mortality decreases with age whereas in old age it increases (Figure 1). In medical terminology infancy refers to new born under one year of age. However, in reality the decrease of the death rate continues until the age of 10, For humans, the increase of the death rate is described by the well-known law of Gompertz [4]. This law can be summarized for by saying that the death rate doubles approximately every 10 years of age. Even the mortality of small animals such as rotifers [3] exhibit similar behaviour as is shown within the inset in Figure 1.

It has been suggested in the literature that non-biological systems obey a similar law however evidence of such behaviour in non-biological systems is not easy to find. Very recently Richmond et al. [5] studied the mortality of systems consisting of soap films and confirmed the bathtub nature of such systems. However, the systems were relatively small and towards the end of life, whilst the mortality increased there was no clear evidence of Gompertz behaviour. In this paper we present evidence for company mortality which mirrors the behaviour shown in Figure 1. The mortality of start-up companies decreases according to a hyperbolic law whereas the mortality of mature companies increases and the long-term trend is in accordance with the Gompertz law. This is shown in the next section. In Section 3, we present a simple model with offers and explanation as to why such behaviour can be expected for complex systems. We close with comments and thoughts for further studies.

## 2. The Mortality of Companies

### 2.1. Start-Up

Much has been written about the survival of start-up companies. Usually this is directed to reasons why such companies fail and do not manage to survive the so-called valley of death in which companies fail due to inadequate working capital. Many other reasons can lead to failure, poor management, marketing, etc. Here, we are not concerned with these micro details rather we shall explore the mortality from a physics perspective looking for general features which characterize the mortality of all companies. For our purposes, a useful dataset is provided at LinkedIn in a paper by McIntyre [6]. Here, can be found survival data for cohorts of companies from their start-up year of 1994 through to 2021. More data is provided for similar cohorts beginning in 1995 and all years through to 2020. Each dataset consisted of over 500,000 companies ensuring good statistics. Earlier data for the period 1947–1954 is given by Steidl [7]. Steidl differentiates between manufacturing, retail and service industries. We show in Figure 2, survival probabilities for both data sets. The broad trend is similar but clearly the data for 1947–1954 falls more steeply than that for more recent years.

From this data for the survival probability, *σ*(*t*), we can compute numerically the ‘force of mortality’, or more simply the mortality, *μ*(*t*). By definition this is the conditional probability that given a person is alive at time *t*, they will die in within the time interval [*t*, *t* + Δ*t*]. It is equivalent to the rate of death conditional on life at time *t*. It follows from this formal definition that it is equal to the ratio of the unconditional survival probability density and the survival probability at time *t*:$$\mu \left(t\right)=-\frac{1}{\sigma \left(t\right)}\frac{\Delta \sigma \left(t\right)}{\Delta t}$$

The Steindl data is fitted extremely well by a hyperbolic function as can be seen on Figure 3. For the average values we find $\sigma \left(t\right)=0.60t^{-0.48}$. This allows us to compute the mortality directly by simple differentiation. Thus $\mu =0.48t^{-1.0}$ However, from the figure it is readily seen that neither an exponential nor a hyperbolic function fits the McIntyre data and a simple first order difference procedure was used to compute annual values for the mortality which does however follow a hyperbolic function over much of the timescale. Thus, both McIntyre and Steindl data sets decay in a similar way following closely the hyperbolic trend $y=A/x^{\gamma }$. Decaying with a power law of −1, the Steindl data follow the value observed for human mortality. The more recent data decays more slowly.

However, it may be seen in Figure 3 that for the recent McIntyre data there is hint of a minimum in the mortality versus time after approximately 20 years which is roughly the same as the minimum observed for humans. Such a minimum is not yet evident in the data of Steindl which only available for up to 10 years.

### 2.2. Mortality of Mature Companies

Having easy access to the UK FTSE100 index we chose to begin here. Comparing the composition for the FTSE 100 when it was first established in 1984 with that in 2021 we can establish 53 companies missing from the 2021 index. The company pages on Wikipedia then provides dates for both birth and death. 

At this point a word of caution is in order. Within this list some companies did die in the sense of going bankrupt. However, others were taken over or merged into another company. Here, we did not differentiate between these different modes of ‘death’. Takeovers and mergers were simply regarded as a point of death. Clearly a takeover or merger ‘deaths is different in nature to a simple bankruptcy. In a sense such a death may not be dissimilar to deaths which occur in some biological systems such as that of a caterpillar as it becomes a butterfly. However, our dataset here is small and we leave further investigation of this point for another study for which a larger index or examination of multiple indices is required.

The lifetime of our 53 companies varies from 13 to 259 years. The one with the shortest lifetime is an oil company; the longest is a brewery. In between we see many types of company. For example: food production, electronics and telephone companies, banks and investment trusts. Figure 4 shows the survival probability of the 53 companies. This was computed simply using the data sets. 

Unlike the data for early-stage companies, the survival probability shown in Figure 4 for the set of mature companies is clearly not smooth. Applying the route used previously to compute the mortality leads to a result which exhibits a number of anomalous sharp peaks which remain despite extensive smoothing of the data. One of us (PR) is grateful to an anonymous referee for pointing out the folly of this procedure. To work around this problem we followed a different procedure. From the data for the survival probability, we first computed the negative of the logarithm. Numerical derivatives were then computed from the resulting data using the central difference approximation yielding the mortality. This procedure has the added benefit of avoiding the numerical division by the survival probability, thus:$$\mu^{i}=-\frac{Ln\sigma^{i+1}-Ln\sigma^{i-1}}{t^{i+1}-t^{i-1}}$$

The result is shown in Figure 5. The dots are computed data points and whilst there is scatter, the solid curve which an exponential fit corresponding to Gompertz behaviour fits reasonably well.

Finally, Figure 6 shows both sets of data (start-up and mature companies together in the manner of Figure 1 for human mortality.) The left-hand curve for small companies is plotted on a log-log scale; the mature company data is shown on a log-linear scale. The ordinate scales are identical. 

The general trend follows that for human mortality shown in Figure 1. Start-up or early life mortality falls in a hyperbolic fashion; mature mortality trends upwards in the manner of Gompertz. From Figure 3, we have noted that the minimum for the more recent start-up company data seems to be around 25 years. However, from the figure it is clear there is a gap between where the deathrate appears to rise (~20–30 years) for the early-stage companies and the level at which mature companies has reached at the same age. However, the earlier data from Steindl falls more steeply and assuming no change in the trend the gap could be better closed with a minimum between the mortalities of early stage and mature companies of around 30–40 years. Why might the McIntyre data be so different? We know from studies of human mortality that data from different time eras can behave in this way. For example, modern medicine reduced substantially the mortality for babies with congenital defects. Here, we have two data sets for small companies taken from quite different time eras. The period 1947–1954 was a period of reconstruction after World War 2 and the nature of small companies then depended on large amounts of capital investment as indeed had been the case since the industrial revolution. However, with the advent of modern computers, the situation changed. Since the 1990s it has been to start up a company with little capital being dependent more on knowledge and computers than intensive amounts of capital. Microsoft for example was set up by Bill Gates in his garage and Google began as an undergraduate project. Using a biological term, we might say we are comparing two different species of company before and after the 1990. We would see similar discrepancies comparing say, human infant mortality with the mortality of adult elephants. Ideally, we should have data for a cohort of similar companies which have evolved in similar environments. Our FTSE100 data is taken over an era extending from the late 20th century back to the 17th century. Therefore, it seems not unreasonable to compare against this mature data with the earlier Steindl data than the McIntyre data. Perhaps even earlier data for the start-up companies might trend even more steeply downwards. A much larger group of similar mature companies might be collected from US data. The S&P 500 perhaps although the time period will be more limited going back perhaps only to the middle of the 19th century. More time needs to elapse before we shall see sufficient data for mature companies to compare with the McIntyre data.

An interesting point is that, whereas for humans medical advances have led to a decrease in mortality, for companies, it seem over time the mortality of early stage companies has increased. The opportunity to set up a company with little capital makes it easier to begin a business, but then perhaps it is also easier too to stop trading. Finally, we note in passing that based on the data we have and extrapolating the trend beyond the maximum data point to where it reaches a value of unity, the results predict a maximum company life time of 283 years. This assumes no takeovers or mergers—which we have seen is not the case. Nevertheless, it will be interesting—for others!—to see if this outcome holds in the modern world.

## 3. The Minmax Model of Mortality

To explain the different forms of early and mature life mortality Richmond and Roehner offered a ‘MinMax’ model where the system was decomposed into elements each of which could function correctly or fail in a random way. For full details we refer the reader to the publication [1]. Here, for completeness we summarize the idea and results. In the case of the human these various elements could be thought of as the different organs (for example: heart lung, brain, etc.). For companies we might think of various departments or functions of the company such as marketing, finance, production etc.) such as shown schematically in Figure 7.

We need a way to describe mathematically whether each element as well as the whole is functioning effectively or not. For simplicity we normalize the life span, Xi, of the elements to [0, 1] where 1 represents the maximum life span of an element. Moreover, all elements are supposed identical.

### 3.1. Mature Mortality

In the model we define the ultimate death of the company to have occurred when all elements of the organization have failed. For the simple 4 element system shown in Figure 8 this may be expressed by saying that if X1 = 0.5, X2 = 0.3, X3 = 0.7, X4 = 0.1 then the age of death is represented by the random variable Z = 0.7, in other words, Z = Max(X1, X2, X3, X4). In [1] we give the complete derivation of the density function *f_p_*(*x*) for *p* elements in terms of the density *f*(*x*) and cumulative distributions *F*(*x*) for the single elements:$$f_{M,p}\left(x\right)=pf\left(x\right)F^{p}\left(x\right)$$

For the simple case of a random variable with a uniform density over the interval (0, 1); in this case: for x ∈ (0,1): *f*(*x*) = 1, *F*(*x*) = x. Thus for *x* ∈ (0,1): *fZ*(*x*) = pxp − 1. This function, shown in Figure 8b for *p* = 2, 4, 8, 15, is a power law function that *increases* fast with age. This is consistent with a Weibull distribution but when *p* becomes large it has the shape of an exponential which is qualitatively consistent with Gompertz’s law according to which the probability of death increases exponentially with age.

### 3.2. Early-Stage Company Death

Again using the same ideas, early stage death would mean that the age of death is: W = 0.1, the is W = Min(X1, X2, X3, X4). Again we refer the reader to [1] where it is shown that for *p* elements in terms of the density *f*(*x*) and cumulative distributions *F*(*x*) for the single elements the density function for early death is
$$f_{W,p}=pf\left(x\right){\left[1-F\left(x\right)\right]}^{p-1}{}_{}$$

For the simple model above we see that *f_W,p_*(*x*) = *p*[1 − *x*]*^p^*^−1^. Consequently, the probability of early-stage death, illustrated in Figure 8a is a *decreasing* function of age, consistent with what is expected for infant mortality.

## 4. Discussion and Conclusions

The interesting conclusion is that the broad trend of company mortality mimics that of humans. Perhaps this is not surprising since companies reflect human behaviour and ingenuity. As for human mortality, the minmax model gives some insight into the behaviour of company mortality. Small companies are known to fail as a result of a particular problem: a new product fails to succeed in a market, new finance is not forthcoming or production is found to be problematic. However, large companies that have evolved beyond the early stage can usually compensate for single department problems. Moreover, it would seem from the data that the trend in early-stage mortality is for it to have risen over the years. Could this be due to it being easier for an entrepreneur to set up a business? Moreover, with the need for limited capital investment relative to earlier times, could it be easier to close down a failing business?

From the minmax idea then we can understand the general shape of the mortality curve and as far as we are aware, this work is the first to show this behaviour for a non-biological system. However, why should the minimum occur around the age of 20–30 years in the manner of human mortality? Could this be linked to the complete passing of the first generation of employees over to a new group who are fully able to grapple with management of complexity as opposed to the skills offered by the initial entrepreneurs? More studies with new data sources are needed to explore this in more detail.

## Figures and Tables

**Figure 1 entropy-24-00208-f001:**
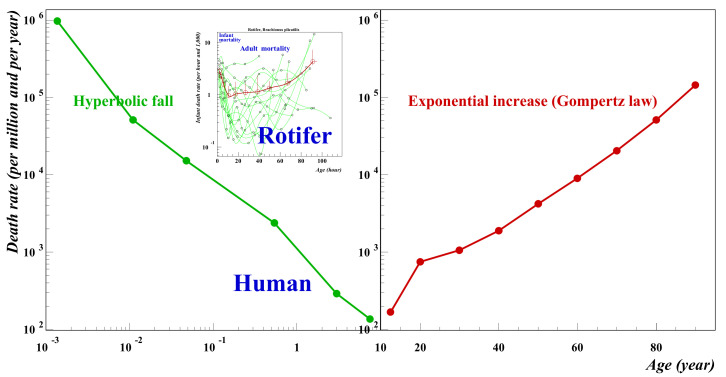
Infant versus old age human mortality. The data are for the US over the period 1999–2016. Between birth and the age of 10 (note the log-log scale) the infant mortality rate falls off as a power law: *μ*(*x*) = A/*x*^γ^ where the exponent γ is 0.99 and usually of the order of 1. After the infant phase comes the aging phase (note the linear-log scale) during which the death rate increases exponentially: *μ*(x) = *μ*(0) exp(α*x*) in agreement with Gompertz’s law and for humans α = 0.079. Source: Wonder-CDC data base for detailed mortality data.

**Figure 2 entropy-24-00208-f002:**
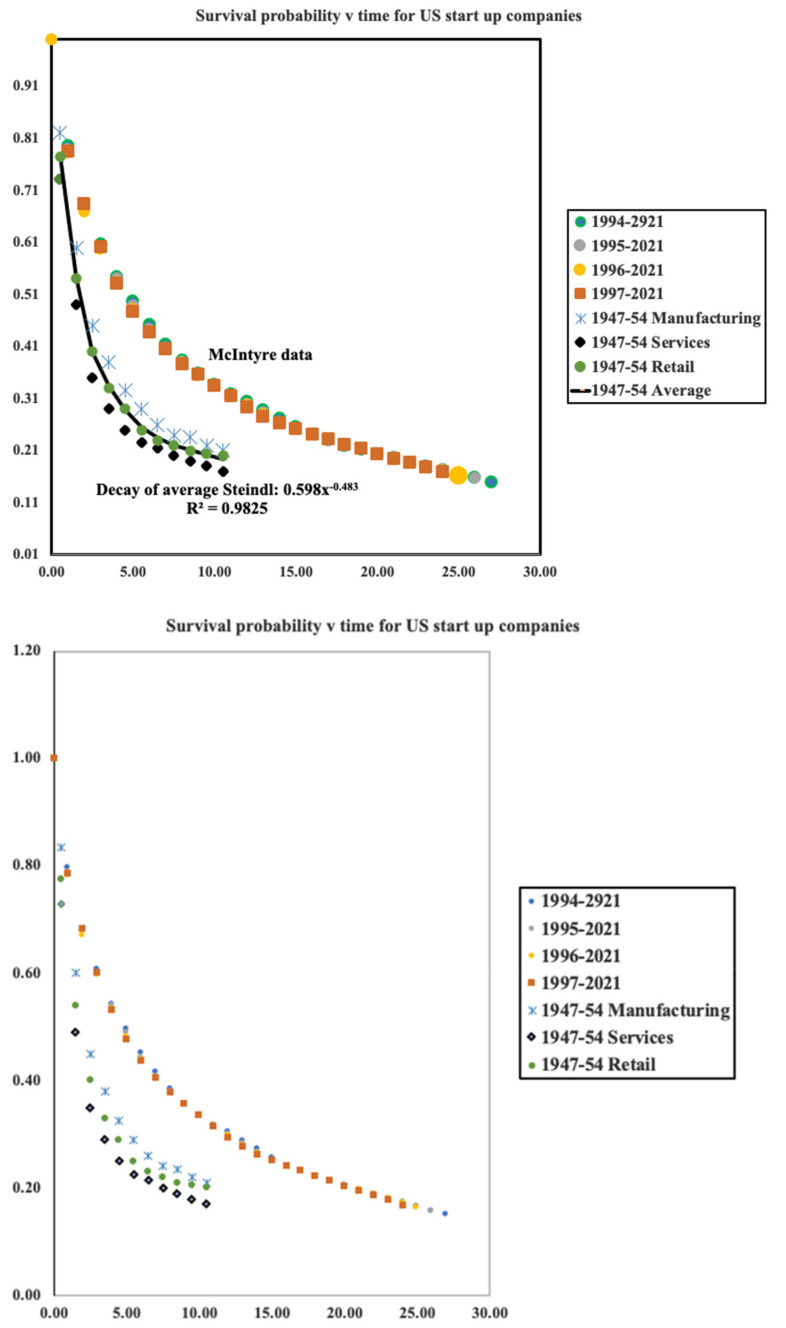
Survival probabilities for US start-up companies over the period 1947–1954 and 1994–2021. Data sources: McIntyre [2] and Steindl [3].

**Figure 3 entropy-24-00208-f003:**
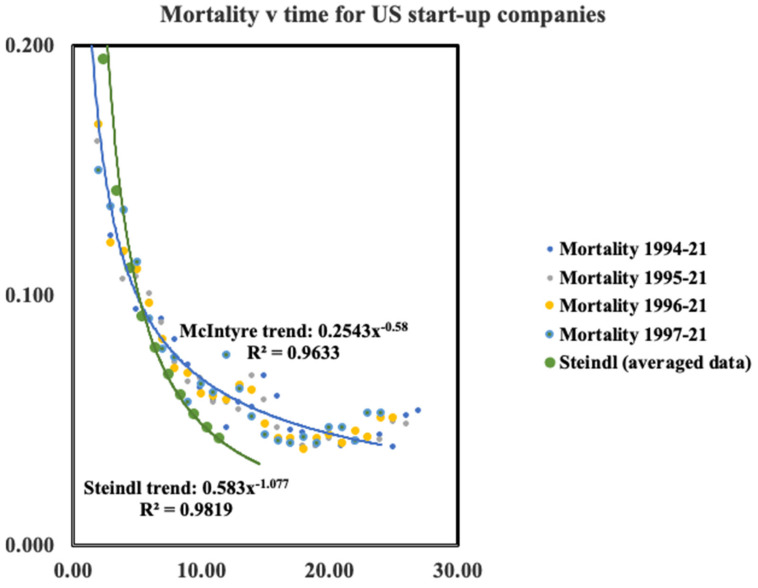
Age specific mortality for US start-up companies using data shown in Figure 2.

**Figure 4 entropy-24-00208-f004:**
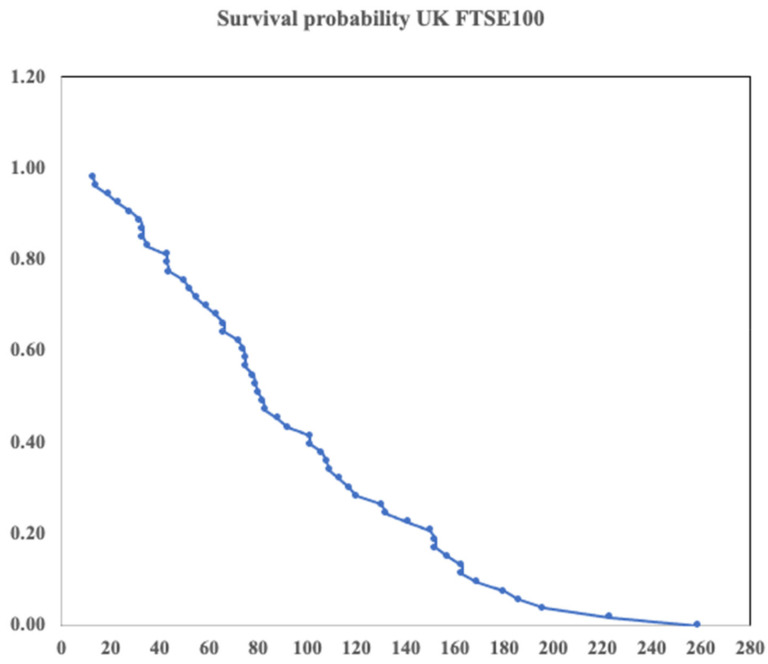
Survival probability for 53 FTSE100 companies which ceased trading between 1921 and 1984, the date the FTSE100 began being compiled.

**Figure 5 entropy-24-00208-f005:**
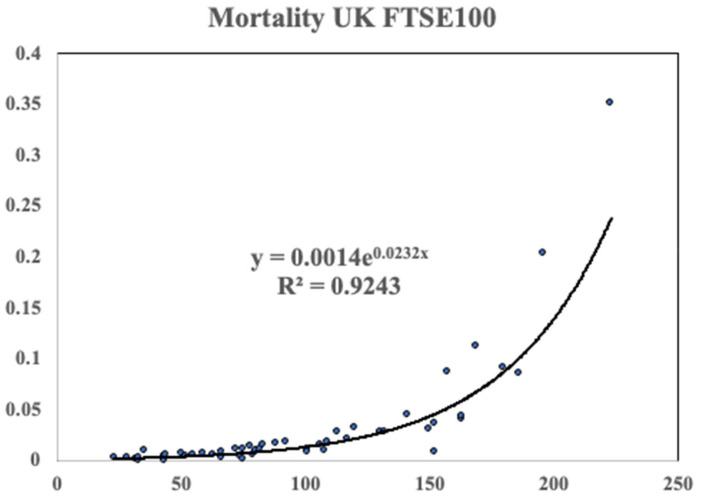
The mortality of the FTSE100 data on a linear plot. The data was computed using the method outlined in the text. The solid line is an exponential fit corresponding to Gompertz like behaviour.

**Figure 6 entropy-24-00208-f006:**
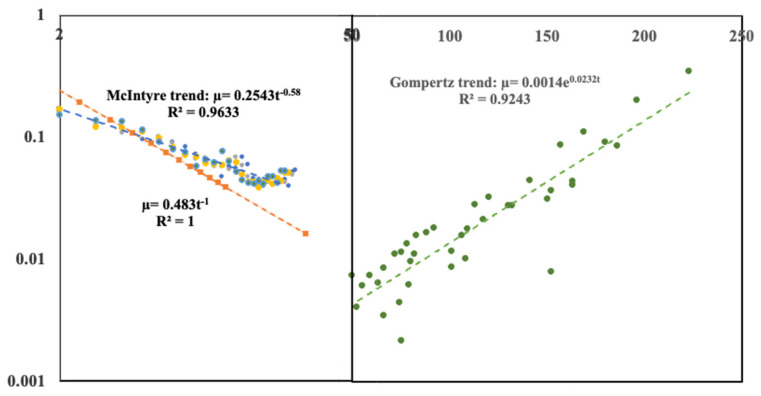
The left-hand graph is the small company mortality data plotted on a log-log plot; the right-hand graph is the mortality data for mature companies plotted on a log-linear plot. The ordinate scales for the mortality or annual deathrate are identical for both data sets. The abscissa for the early-stage companies is a logarithmic and extends to 50 years of age; the abscissa for the mature companies is a linear scale extending from 50 years to 250 years. The solid lines are the same regression fits shown in the earlier figures with details within the insets. However, here we have extended the early stage data trend line further out to around 30 years.

**Figure 7 entropy-24-00208-f007:**
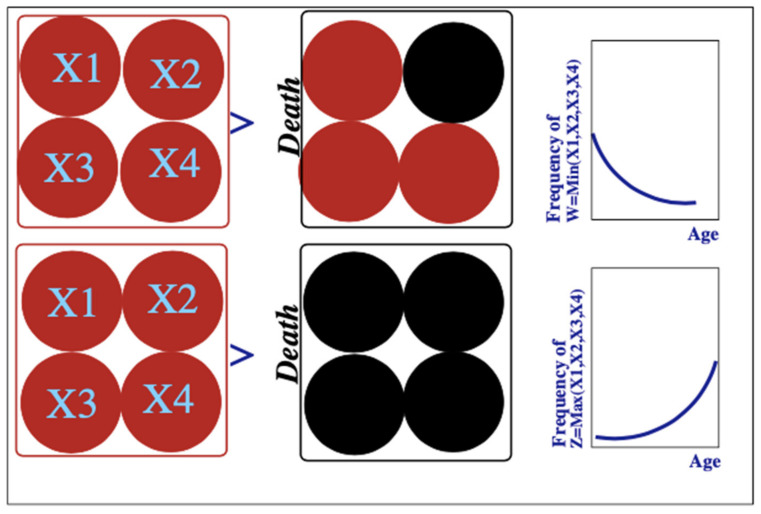
Illustration of the decomposition of an organism into vital organs and the difference between early life and mature life mortality mechanisms. The upper diagrams illustrate early life death. It is the consequence of the failure of a single vital organ. The lower diagrams show a mature death which is a consequence of uniform deterioration of all the vital organs. The graphs on the right-hand side show the implications of these mechanisms in terms of age-specific death rates: decreasing for early life as observed in infant death rates, increasing for mature death as seen in old-age.

**Figure 8 entropy-24-00208-f008:**
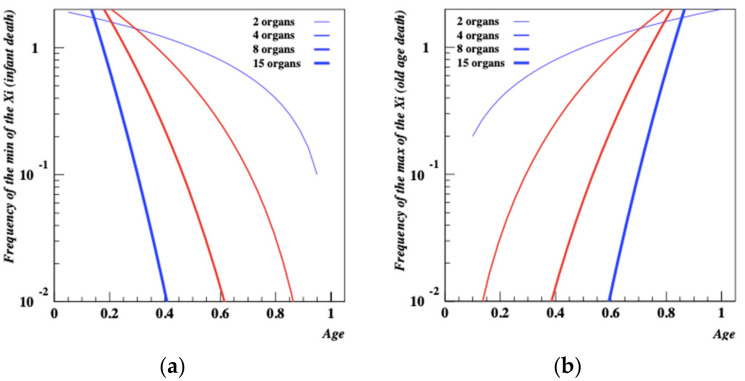
(**a**,**b**) MinMax model density functions for a set of random variables which represent age specific deathrates in early stage and mature companies. When the elemental structure of the company becomes large, deathrates for mature companies become exponential.

## Data Availability

Not applicable.

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
