# Peer review of "On the Mortality of Companies"

_entropy, 2022, doi:10.3390/e24020208_

Round 1

Reviewer 1 Report

I like the scope of the paper and think it is worth publishing. However, I feel the authors have been a little lazy in trying to understand the strange "spikes" that they report on the distribution of lifetimes. For me, this is either just noise coming from the size of the data, or some kind of "resonance" coming from a coincidence of the birth year/death year with some macro effect (for example 1929 or 2007, etc.). So I would urge the authors to do two things: a) a monte carlo simulation of the model to estimate the expected error bars on the data b) for the anomalous peaks, report the start-end dates and check whether there is a clustering around a certain pair of dates.  

Author Response

The authors are gratified to learn that the referee finds the manuscript interesting. We have revisited the calculations and do not accept that the 'spikes' are of a random character. We offer a different suggestion for their origin. From the graph they seem to have a fairly regular period of around 40-50 years. We suggest that in fact they are a manifestation of long wavelength cycles of the type proposed by the Russian economist Kondratieff some years ago. We are not clear why or how a Monte Carlo calculation would shed light on the matter. How would we select the transition probabilities for the system? These are what we are seeking via the empirical data.

Reviewer 2 Report

The methodological proposal to study the survival and mortality of companies in their initial and mature stage is interesting. The authors particularize their analysis to companies from the United States and the United Kingdom. However, the contribution of the article is not clear. In which way this analysis differs from previous studies, especially those carried out by the same authors?

Moreover, the article is written in the draft style and is not fully debugged. The writing is sloppy and has many typographical errors and the language is informal/colloquial. References are scarce, if not null most of the time. The mathematical models are not described properly, so it is not possible to verify the veracity of the conclusions.

An important issue is the 16% (564 words) plagiarism detected by Ithenticate from Richmond, P., & Roehner, B. M. (2021). A joint explanation of infant and old age mortality. Journal of Biological Physics, 1-11. Please rewrite section 3 mainly.

Some particular observations are the following:

Figures are not of adequate quality. Please increase the resolution to 300 dpi. Omit titles and instead describe them precisely in captions.

Equation of line 78 is not explained term by term. It is important to cite the original source or write their derivation. At least should be given an intuition behind the formula.

Line 122: Please describe the excel method or version otherwise it is impossible to comply with the reproducibility principle in science.

Line 130: How can we convince ourselves of this behavior? What are the statistical tests used to verify it?

Line 138: The narrative is too informal.

Line 160: The model MinMax must be explained at least in an intuitive way.

Line 227: Example of typographical error.

Author Response

The authors are pleased that the referee feels the work is interesting. Quite what the author means by 'sloppy writing' we are not sure. However we have completely overhauled the text and hopefully it is now acceptable to the referee. In addition the graphical diagrams have been revisited and redrawn. The trends discernible from the data are given and statistical fits marked on the various graphs. 

Concerning the section which discusses the model we refer the reader to our previous publication (which is the original source) where it is fully developed. We have only included here the ideas behind the model and a summary of the results. But given even this it is not surprising that there is some overlap - 16% is probably not unusual. However the referee should note that the contexts of this paper and our previous paper are totally different. This paper deals with mortality of commercial businesses; the previous paper is concerned with human mortality. The point is that the same model can be used to gain insight into both human and business systems. 

Finally I note that we have revised the method used to derivethe mortality data. The use of binning and the black box tool within MS Excel has been abandoned. A more direct method fully explained in the text has been used.

Round 2

Reviewer 1 Report

>> I am not satisfied with the authors reply. I think a null hypothesis

>> case where such spikes are absent in theory allows one to generate

>> random samples of the same length as the authors' data and estimate

>> error bars. But my second suggestion is I think more useful: focusing

>> on the spikes, can the authors at least report the list of birth

>> years and death years of the companies within the spikes and see

>> whether there is an excess of some special pair of years? IMHO I do

>> not believe that any kind of business cycle idea could explain such

>> sharp peaks.

Author Response

Thank you for the comments. The manuscript has now been extensively revised. Using a different approach explained in the text, the unexpected fluctuations are now no longer present. The mature mortality shows only Gompertz like behaviour. Additional data from an earlier time era has also been introduced which compares more favourably with the mature company data. This illustrates the minimum of mortality at a round 30 years again similar to human mortality data.

Reviewer 2 Report

The authors addressed the observations.

Author Response

Thank you for the comments acknowledged within the text which has now been extensively revised. Using a different approach explained in the text, the unexpected fluctuations are now no longer present. The mature mortality shows only Gompertz like behaviour. Additional data for early startup companies from an earlier time era has also been introduced. Comparing this alongside the mature data which is also from an early time era we can see now more clearly the existence of a minimum of mortality at a round 30 years again similar to human mortality data. 

Round 3

Reviewer 1 Report

The paper can now be accepted, the authors have corrected their big error